# Post-harvest treatment of wild-simulated ginseng under climate-smart environmental conditions

Solhee Kim[1], Taegon Kim[1,2], Jungyeon Kim[3,4]*, Kyo Suh[4,5]*

1 Department of Smart Farm, Jeonbuk National University, Jeonju, Republic of Korea, 2 Institute of Agricultural Science & Technology, Jeonbuk National University, Jeonju, South Korea, 3 Institute of Food Industrialization, Institutes of Green Bioscience and Technology, Seoul National University, Gangwon-do, Republic of Korea, 4 Graduate School of International Agricultural Technology, Seoul National University, Pyeongchang-gun, Gangwon-do, Republic of Korea, 5 Institute of Green Eco Engineering, Institutes of Green Bioscience and Technology, Seoul National University, Gangwon-do, Republic of Korea

* kyosuh@snu.ac.kr (KS); kim131812@snu.ac.kr (JK)

## Abstract

Wild-simulated ginseng (WSG; *Panax ginseng* C.A. Meyer) is prized for its unique ginsenoside profile and medicinal properties; however, traditional cultivation faces challenges, such as declining survival rates and prolonged growth periods. We aimed to develop an enhanced post-harvest treatment using climate-smart conditions to improve survival rates, boost root mass, and preserve the characteristic ginsenoside profile of WSG. Three-year-old WSG roots were transplanted and cultivated in a controlled smart facility in regulated light and irrigation conditions for five months using premium ginseng soil, a certified organic ginseng soil medium. Growth performance was monitored and ginsenoside profiles were analysed via ultra-performance liquid chromatography. Post-treatment, WSG exhibited a 2.5-fold increase in root weight and an overall survival rate of 97%. Total ginsenoside content reached 10.458 mg/g dried ginseng, with notably high levels of Re (7.716 mg/g) and the presence of rare compounds, such as Compound K and Rg3. The root-to-shoot ratio exceeded 1.23, indicating efficient resource allocation. These results demonstrate that climate-smart post-harvest treatment effectively enhances root development and maintains the medicinal quality of WSG, offering a promising strategy to overcome the limitations of conventional cultivation and improve its commercial viability.

## Introduction

Wild-simulated ginseng (WSG; *Panax ginseng* C.A. Meyer) is a perennial herb belonging to the family Araliaceae, naturally growing in East Asian mountain regions [1]. WSG cultivation requires natural mountainous areas without artificial facilities, such as shade structures, using direct seeding or transplanting of nursery plants without pesticides or chemical fertilizers [2]. Although genetically identical to conventional ginseng—which requires a six-year growth period and increased agricultural

**Data availability statement:** All relevant data are within the manuscript and its Supporting information files.

**Funding:** The author(s) received no specific funding for this work.

**Competing interests:** The authors have declared that no competing interests exist.

**Abbreviations:** WSG, wild-simulated ginseng; FCG, field-cultivated ginseng; DG, dried ginseng; PPT, protopanaxatriol; CK, Compound K.

chemical inputs after the third year—WSG grows naturally for 7 to over 20 years [3]. This fundamental difference in cultivation methods, especially in growth duration and chemical use, distinguishes WSG from field-cultivated ginseng (FCG).

WSG exhibits unique ginsenoside profiles and age-dependent concentration increases, along with distinctive compounds not detected in FCG [4,5]. Several studies have demonstrated that WSG synthesizes a wider array of ginsenosides than that by FCG, primarily due to environmental stresses in its natural habitat [2]. These stresses activate secondary metabolism, producing a range of bioactive compounds with health benefits [6]. This broader spectrum of ginsenosides enhances its therapeutic properties, including anti-cancer [7–9], anti-inflammatory [10,11], and immuno-modulatory effects [12].

Recent societal changes, including aging population in Korea and increased health consciousness following the COVID-19 pandemic, have driven growing demand for WSG. By 2023, WSG was cultivated on 13,858 hectares by 3,792 forestry house-holds [13]. Annual production of 254,111 kg yields a market value of 62.9 billion won, with prices up to 250,000 won per kg. These figures position WSG as one of the most valuable forest products of Korea, reflecting its premium market status.

Although WSG can be certified for sale after three years of growth, its intermediate root size at this stage affects market acceptance, with trade favouring roots aged at least seven years [14]. However, Korea Forest Service statistics show that WSG survival rates drastically decline with cultivation duration (Korea Forest Service, 2024). The average survival rate is 77.8% for one-year-old roots, decreasing to 55.2% at three years and 38.8% at five years. For seven-year-old roots, which are in high demand, the rate declines to about 25.3%. This trend reflects an annual loss rate of 15–20%, with the largest decrease (16.5%) occurring from four to five years.

Here, we aimed to develop an enhanced post-harvest treatment using climate-smart conditions to improve survival rates, boost root mass, and preserve the characteristic ginsenoside profile of WSG. We employed a controlled system that manages temperature, humidity, and light for WSG harvested during dormancy (3–4 years), applying a short-term (up to six months) post-harvest treatment in a controlled soil environment.

## Materials and methods

### Cultivation in smart facility for post-treated WSG

We obtained certified three-year-old WSG plants from the Pyeongchang Wild Ginseng Association (Gangwon-do, Republic of Korea). We transplanted the plants on April 20, 2022 and cultivated them for five months in the smart facility with environmental controls for light, irrigation, and temperature regulation. Environmental conditions were based on known physiological thresholds for WSG; the temperature was maintained below 21°C and light intensity under 200 PPFD (≈5,000 lux) to prevent photoinhibition and stress-induced growth inhibition [15,16]. Humidity was not actively regulated but remained stable under shaded indoor conditions. We used premium ginseng soil (Shinsung Mineral Co., Ltd., Cheongan-myeon, Republic of

Korea)—the first certified organic ginseng cultivation medium in Korea (Certification No. 1-2-109)—as the post-harvest treatment medium. Further details on cultivation techniques, environmental management, and container design are provided in S1 Appendix.

## Ginsenoside analysis

We analysed nine major ginsenosides (Rb1, Rg1, Rg3, Rd, Re, Rf, Mc1, F2, and Compound K) using the Waters ACQUITY® ultra-performance liquid chromatography (UPLC) system (Waters Corp., Milford, MA, USA). We operated the instrument at a gas temperature of 300 °C and a flow rate of 5 L/min. The nebulizer pressure was maintained at 45 psi, while the sheath gas temperature and flow rate were set at 250 °C and 11 L/min, respectively. Both positive and negative modes were employed, with a capillary voltage of 3,500 V and a nozzle voltage of 500 V. To ensure consistency with previous studies and eliminate moisture-related variations, we used dried post-treated WSG samples for all analyses, allowing direct comparison with existing data on both WSG and FCG.

## Impact of post-treatment protocol on WSG growth and survival characteristics

To evaluate post-treated WSG growth patterns, we monitored root weight changes over five months. Specimens were initially subjected to visual screening for preliminary separation, then weighed to assign them into two groups based on a 1.0 g fresh weight threshold: Group A (> 1.0 g; mean: 1.58 ± 0.21 g) and Group B (≤ 1.0 g; mean: 0.74 ± 0.18 g). This grouping criterion was informed by a bimodal weight distribution observed across 220 specimens (Fig 1A). The final harvest at five months revealed substantial growth in both groups: Group A achieved an average root weight of 3.85 g (a 2.44-fold increase), whereas Group B reached an average of 1.53 g (a 2.07-fold increase) (Fig 1B).

To evaluate the effect of post-treatment on the ginsenoside profiles of WSG, UPLC analysis was performed. The total ginsenoside content of post-treated WSG was 10.458 mg/g of dried wild-simulated ginseng (DG) weight and considering an average root weight of 4.0 g, each root contained approximately 41.83 mg of total ginsenosides (Table 1). Of the analysed ginsenosides, Re was the most abundant (7.716 mg/g DG), followed by Rb1 (1.293 mg/g DG) and Rg1 (1.195 mg/g

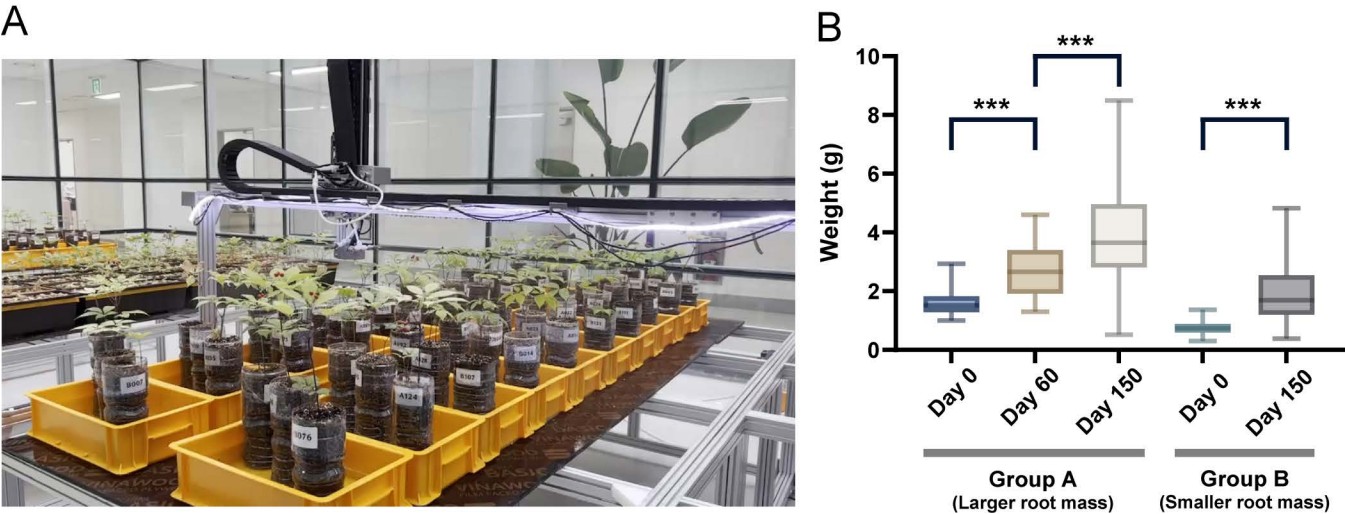

**Fig 1. Post-harvest treatment of wild-simulated ginseng (WSG).** (A) Smart facility for post-treated WSG at Pyeongchang campus in Seoul National University. (B) Changes in average root weight of post-treated WSG over the treatment period. Group A (initial n = 220) and Group B (initial n = 220) represent larger and smaller roots, respectively. Samples (n = 40) were excluded at 2 months to measure the ratio of underground to shoot fresh weight. The survival rate was 96.8% for Group A and 97.3% for Group B. *** represents *p* value < 0.0001 (*t* test; FDR adjusted).

DG). Ginsenoside compounds K and Rg3, which are known to be present in WSG, however, absent in FCG, were detected in post-treated WSG [5].

The post-treatment protocol successfully addresses a key challenge in WSG cultivation—maintaining ginsenoside profiles while achieving greater root mass and survival rate. Excluding the 40 specimens harvested at two months from Group A, only seven specimens in Group A and six in Group B were lost during the treatment period, resulting in an estimated survival rate of 97%. This high survival rate, combined with the substantial weight gains observed in both groups, suggests that the post-treatment protocol effectively supported root development while maintaining plant viability. These results indicate that post-treatment conditions promoted proportional growth regardless of initial root size, as both groups exhibited similar relative weight increases over the five months. The enhanced root development observed in our study (a 2.5-fold increase over five months) considerably exceeds the growth rates reported in traditional WSG cultivation, where annual root weight increases typically range from 30 to 50% [17]. The post-treated WSG also contained notable amounts of ginsenosides, such as CK and Rg3 [5], which are significantly more abundant in WSG than in FCG (Table 1). This suggests that post-treatment enhanced size and maintained high survival while preserving the distinctive characteristics of WSG.

Pre-treatment ginsenoside profiles were not analyzed in this study due to the study design, which prioritized monitoring of post-treatment growth and biochemical outcomes. As a result, absolute changes in ginsenoside content before and after treatment could not be quantified. Nevertheless, the detection of rare ginsenosides such as Compound K and Rg3—compounds typically absent in three-year-old FCG—may indicate a potential induction of secondary metabolite biosynthesis under climate-smart post-harvest conditions. This interpretation remains speculative in the absence of baseline values, and future studies should include paired pre- and post-treatment ginsenoside analyses to validate the observed biochemical trends.

In addition, the long-term reusability of the certified organic soil used in this study was not evaluated. Future investigations should assess its structural integrity, nutrient retention, and microbial stability across multiple planting cycles to determine its applicability for repeated use in commercial WSG production systems.

## Conclusions

This study demonstrates that post-harvest treatment of WSG under controlled environmental conditions enhances root development while preserving its distinctive medicinal properties through systematic management of environmental

**Table 1. Analysis of ginsenoside content using UPLC in post-treated WSG.**

| Ginsenoside | | Post-treated WSG (Unit: mg/g, DG) |
|---|---|---|
| PPD | Rb1 | 1.293±0.024 |
| | Rg3 | 0.017±0.001 |
| | Rd | 0.218±0.017 |
| | Mc1 | 0.002±0.000 |
| | F2 | 0.011±0.001 |
| | CK | 0.006±0.001 |
| PPT | Rg1 | 1.195±0.121 |
| | Re | 7.716±0.257 |
| | Rf | 0.001±0.000 |
| Total | | 10.458±0.421 |

Data represent mean values±standard error (mg/g). DG: weight of dried wild-simulated ginseng.

factors. Key findings include a 2.5-fold weight increase over five months, survival rates exceeding 96%, and retention of the characteristic ginsenoside profiles.

The post-treatment protocol addresses several crucial challenges in traditional WSG cultivation. Offering controlled post-harvest conditions that preserve the unique properties of WSG may resolve the issues of the industry with unstable production and variable quality. The preservation of rare ginsenosides, such as Compound K and Rg3, confirms that this approach maintains the medicinal value characteristic of WSG while improving production efficiency. The post-treated WSG evaluated in this study occupies a distinct position between conventional field-cultivated and wild-simulated ginseng. It shows improved root growth and preservation of pharmacologically relevant ginsenosides under a shorter cultivation window. While the production cost is expected to be higher than that of FCG, it remains significantly lower than that of long-term WSG cultivation. This balance of cost and efficacy may offer a practical and scalable alternative for premium ginseng markets.

Achieving significant growth within five months, high survival rates, and desirable pharmacological properties suggests potential economic benefits. This method could improve the commercial value of previously unmarketable WSG while ensuring consistent quality standards.

Future research should investigate treatment conditions for different genetic stocks and seasonal effects. This research represents a step toward overcoming current limitations in WSG cultivation while preserving its valuable medicinal properties.

## Supporting information

**S1 Appendix. Cultivation techniques, environmental management, and container design.**
(DOCX)

**S1 Fig. Changes in chlorophyll fluorescence (Fv/Fm) of WSG leaves under different light and irrigation conditions over a 15-day period.** Light/$H_2O$: natural light with water, Light/HOCl: natural light with HOCl, Dark/$H_2O$: shaded with water, Dark/HOCl: shaded with HOCl. Measurements were taken at 1, 5, 8, 12, and 15 days after treatment. Vertical bars indicate standard errors.
(DOCX)

**S2 Fig. Comparison of shoot development under different light and irrigation conditions.** (A) Three petioles from plants grown under light and dark conditions. (B) Representative leaf morphology of middle leaflets under different light and irrigation treatments (Light/$H_2O$, Light/HOCl, Dark/$H_2O$, Dark/HOCl). Scale bar = 5 cm.
(DOCX)

**S3 Fig. The average weight of WSG roots under different light and irrigation conditions after two months of treatment.** The initial weight of transplanted roots was 1.70 ± 0.40 g (mean ± SD, n = 39). Bars represent mean values ± standard errors (n.s.: not significant, ***: $p < 0.001$, ****: $p < 0.0001$).
(DOCX)

## Acknowledgments

We sincerely appreciate the invaluable assistance of the Green Bio Research Facility Center at the Institutes of Green Bio Science & Technology with the ginsenoside analysis.

## Author contributions

**Data curation:** Solhee Kim.

**Formal analysis:** Solhee Kim, Taegon Kim.

**Investigation:** Taegon Kim.

**Methodology:** Solhee Kim.

**Supervision:** Jungyeon Kim, Kyo Suh.

**Validation:** Solhee Kim.

**Visualization:** Solhee Kim.

**Writing – original draft:** Solhee Kim.

**Writing – review & editing:** Jungyeon Kim, Kyo Suh.

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
