## [Decision Letter · Decision Letter 0]

Dear Dr. Kim,

Thank you for submitting your manuscript to PLOS ONE. After careful consideration, we feel that it has merit but does not fully meet PLOS ONE’s publication criteria as it currently stands. Therefore, we invite you to submit a revised version of the manuscript that addresses the points raised during the review process.

We look forward to receiving your revised manuscript.

Kind regards,

Chun-Hua Wang

Academic Editor

PLOS ONE

Journal Requirements:

Reviewers' comments:

Reviewer's Responses to Questions

**Comments to the Author**

1. Is the manuscript technically sound, and do the data support the conclusions?

Reviewer #1: Yes

Reviewer #2: Yes

2. Has the statistical analysis been performed appropriately and rigorously?

Reviewer #1: Yes

Reviewer #2: Yes

3. Have the authors made all data underlying the findings in their manuscript fully available?

Reviewer #1: Yes

Reviewer #2: Yes

4. Is the manuscript presented in an intelligible fashion and written in standard English?

Reviewer #1: Yes

Reviewer #2: Yes

Reviewer #1: This manuscript presents a valuable study that proposes a post-cultivation treatment strategy for wild-simulated ginseng (WSG) under climate-smart conditions. The approach aims to enhance the added value of WSG by improving survival rates, increasing root biomass, and maintaining a characteristic ginsenoside profile. While the study is significant, the following revisions are recommended to enhance clarity and scientific rigor:

1. Clarification of Grouping Criteria: The basis for dividing WSG into two groups requires further explanation. The use of subjective descriptors such as "large" and "small" is ambiguous and should be replaced with clearly defined, quantifiable parameters (e.g., weight range, root diameter).

2. Terminology Concerning Plant Age: Referring to three-year-old WSG as "seedlings" appears to be inappropriate. It is suggested that the authors use terminology that accurately reflects the developmental stage of the plants based on established ginseng cultivation standards.

3. Inclusion of Pre-treatment Ginsenoside Data: Table 1 presents only the ginsenoside content following post-treatment. To allow for meaningful interpretation of the treatment’s effect, it is essential to also include the ginsenoside profile prior to treatment.

4. Accuracy of Initial Root Data in Figure S4: The description of the initial WSG root weight as "more than 1 g" in Figure S4 lacks precision. Providing exact weight measurements or a statistical summary (e.g., mean ± SD) would improve the credibility and reproducibility of the data.

Reviewer #2: The manuscript cultivates an artificial intelligence climate management system for WSG, which improves survival rate and shortens growth period while ensuring effective ingredient content. I hope the following information can be added to the article:

How are the optimal temperatures and humidity for WSG growth determined?

How many times can organic nutrient soil be planted?

Is there any difference in morphology between artificially cultivated and wild WSGs? How much is the cost difference?

**Do you want your identity to be public for this peer review?** For information about this choice, including consent withdrawal, please see our Privacy Policy

Reviewer #1: No

Reviewer #2: No

---

## [Author Response · Author response to Decision Letter 1]

26 May 2025

Responses to Reviewers

Many thanks to the editors and reviewers for their constructive comments that will help improve our manuscript. Each comment was revised and the contents were highlighted with red colour. The responses to individual comments are in Italics below:

Reviewer #1:

This manuscript presents a valuable study that proposes a post-cultivation treatment strategy for wild-simulated ginseng (WSG) under climate-smart conditions. The approach aims to enhance the added value of WSG by improving survival rates, increasing root biomass, and maintaining a characteristic ginsenoside profile. While the study is significant, the following revisions are recommended to enhance clarity and scientific rigor:

1. Clarification of Grouping Criteria: The basis for dividing WSG into two groups requires further explanation. The use of subjective descriptors such as "large" and "small" is ambiguous and should be replaced with clearly defined, quantifiable parameters (e.g., weight range, root diameter).

We thank the reviewer for pointing out the need to clarify the grouping criteria for WSG specimens. In response, we have revised the manuscript to remove subjective descriptors such as “large” and “small,” and instead provide an objective, quantifiable grouping based on root weight.

Specifically, specimens were initially screened visually to facilitate preliminary sorting, then weighed and divided using a 1.0 g fresh weight threshold. This value was chosen based on the bimodal weight distribution observed in the 220 initial samples, which revealed a natural separation near 1.0 g. Group A included specimens >1.0 g (mean: 1.58 ± 0.21 g), and Group B included those ≤1.0 g (mean: 0.74 ± 0.18 g). These changes have been incorporated into the revised manuscript as follows: Revised text in Page 4, Lines 89-92:

“Specimens were initially subjected to visual screening for preliminary separation, then weighed to assign them into two groups based on a 1.0 g fresh weight threshold: Group A (> 1.0 g; mean: 1.58 ± 0.21 g) and Group B (≤ 1.0 g; mean: 0.74 ± 0.18 g). This grouping criterion was informed by a bimodal weight distribution observed across 220 specimens (Fig. 1A). …”

2. Terminology Concerning Plant Age: Referring to three-year-old WSG as "seedlings" appears to be inappropriate. It is suggested that the authors use terminology that accurately reflects the developmental stage of the plants based on established ginseng cultivation standards.

Referring to three-year-old WSG as “seedlings” was inaccurate and has been corrected in the revised manuscript. Based on established cultivation standards, this developmental stage exceeds the seedling phase. We have therefore replaced the term with “three-year-old wild-simulated ginseng plants”, which more accurately describes the biological maturity of the material used in this study.

“WSG cultivation requires natural mountainous areas without artificial facilities, …, using direct seeding or transplanting of nursery plants without pesticides or chemical fertilizers [2]. ” : Revised text in Page 3, Lines 36.

“We obtained certified three-year-old WSG plants from the Pyeongchang Wild Ginseng Association (Gangwon-do, Republic of Korea). We transplanted the plants on April 20, …” : Revised text in Page 4, Lines 67-68.

3. Inclusion of Pre-treatment Ginsenoside Data: Table 1 presents only the ginsenoside content following post-treatment. To allow for meaningful interpretation of the treatment’s effect, it is essential to also include the ginsenoside profile prior to treatment.

We appreciate the reviewer’s observation regarding the need for pre-treatment ginsenoside data to enable direct evaluation of the treatment’s biochemical effects. In this study, however, ginsenoside profiling was conducted only after treatment, and baseline ginsenoside levels were not measured. This decision reflected the study design, which prioritized post-treatment root growth tracking and chemical composition assessment rather than pre-post biochemical comparison.

To address this limitation, we have added a clarifying statement in the Discussion section that explains the absence of pre-treatment data, and we acknowledge that this limits our ability to quantify biochemical changes induced by the treatment. However, we also note the detection of rare ginsenosides such as Compound K and Rg3—compounds typically absent in three-year-old field-cultivated ginseng (FCG)—which may suggest a positive effect of the treatment on secondary metabolite biosynthesis. Added text in Discussion (Page 6, Lines 118-125):

“Pre-treatment ginsenoside profiles were not analyzed in this study due to the study design, which prioritized monitoring of post-treatment growth and biochemical outcomes. As a result, absolute changes in ginsenoside content before and after treatment could not be quantified. Nevertheless, the detection of rare ginsenosides such as Compound K and Rg3—compounds typically absent in three-year-old field-cultivated ginseng (FCG)—may indicate a potential induction of secondary metabolite biosynthesis under climate-smart post-harvest conditions. This interpretation remains speculative in the absence of baseline values, and future studies should include paired pre- and post-treatment ginsenoside analyses to validate the observed biochemical trends.”

4. Accuracy of Initial Root Data in Figure S4: The description of the initial WSG root weight as "more than 1 g" in Figure S4 lacks precision. Providing exact weight measurements or a statistical summary (e.g., mean ± SD) would improve the credibility and reproducibility of the data.

We have revised the caption of Figure S4 to include the actual initial root weight statistics used in the short-term treatment experiment. Specifically, the initial weight of transplanted roots was 1.70 ± 0.40 g (mean ± SD, n = 39), replacing the previously vague “more than 1 g” description and caption. Revised SI text in Page 10:

“The experiment involved three-year-old or older wild-simulated ginseng roots with a mean initial weight of 1.70 ± 0.40 g (mean ± SD, n = 39). Measurements were taken two months after transplantation.”

“Figure S4. The average weight of WSG roots under different light and irrigation conditions after two months of treatment. The initial weight of transplanted roots was 1.70 ± 0.40 g (mean ± SD, n = 39). Bars represent mean values ± standard errors (n.s.: not significant, ***: p < 0.001, ****: p < 0.0001).”

Reviewer #2:

The manuscript cultivates an artificial intelligence climate management system for WSG, which improves survival rate and shortens growth period while ensuring effective ingredient content. I hope the following information can be added to the article:

1. How are the optimal temperatures and humidity for WSG growth determined?

We thank the reviewer for raising this important point regarding the basis for environmental control during post-harvest treatment. In this study, temperature and humidity conditions were not set based on a single defined "optimal value" derived from empirical testing, but were instead informed by known physiological characteristics of WSG and agronomic standards reported in the literature.

Specifically, our Supplementary Information (Section S2.3) describes the rationale for environmental control:

- WSG is highly sensitive to high temperatures, with light saturation levels dropping sharply above 30°C.

- Therefore, we maintained air and soil temperatures below 21°C, and light intensity below 200 PPFD (approx. 5,000 lux), using natural light and polyethylene shade netting.

- Soil moisture was closely monitored and managed through a combination of bottom-watering and mist irrigation, adjusted seasonally to maintain appropriate substrate conditions.

- While humidity was not mechanically controlled, the shaded, naturally cool indoor conditions of the treatment facility helped maintain stable relative humidity in the range suitable for ginseng post-harvest handling.

This protocol was guided by previously reported environmental thresholds (e.g., Kim et al., 2019) and standard WSG practices recommended by Korean forestry and medicinal plant research institutes. The goal was to provide a low-stress environment favorable for root biomass retention and survival during the dormancy transition, rather than to experimentally define an optimal climate envelope.

“… Environmental conditions were based on known physiological thresholds for WSG; the temperature was maintained below 21℃ and light intensity under 200 PPFD (≈5,000 lux) to prevent photoinhibition and stress-induced growth inhibition [1-3]. Humidity was not actively regulated but remained stable under shaded indoor conditions. … Further details on cultivation techniques, environmental management, and container design are provided in Supplementary Sections S1 and S2.” Added text in Materials and Methods (Page 4, Lines 70-77).

“These conditions were selected based on the known physiological sensitivities of Panax ginseng. Elevated temperatures and high light intensity are known to impair photosynthetic efficiency and reduce biomass accumulation due to photoinhibition and increased respiration [4]. Prior studies have shown that ginseng exhibits optimal performance when light intensity is kept below 200 PPFD, particularly under cooler conditions [5, 6]. While humidity was not mechanically controlled, the shaded and enclosed indoor environment provided moderate and consistent humidity levels comparable to those in natural understory habitats.” Revised SI text in Page 7.

2. How many times can organic nutrient soil be planted?

We acknowledge the reviewer’s interest in the long-term reusability of the certified organic nutrient soil used in this study. However, we did not evaluate repeated planting cycles using the same soil in this experiment, as our focus was limited to the short-term post-harvest treatment and physiological response of WSG.

We agree that the reusability of the soil medium—particularly its physicochemical stability and nutrient retention over multiple planting cycles—is a critical topic for practical implementation. We have added a statement in the Discussion section identifying this aspect as a key area for future investigation. Added text in Discussion (Page 6-7, Lines 126-129):

“In addition, the long-term reusability of the certified organic soil used in this study was not evaluated. Future investigations should assess its structural integrity, nutrient retention, and microbial stability across multiple planting cycles to determine its applicability for repeated use in commercial WSG production systems.”

3. Is there any difference in morphology between artificially cultivated and wild WSGs? How much is the cost difference?

Thank you for raising this important question regarding the morphological and cost-related differences between artificially cultivated and wild-simulated ginseng (WSG).

Regarding morphology, this study did not include a direct experimental comparison between wild-grown and artificially cultivated WSG. All samples were sourced from the same origin and treated under post-harvest conditions to assess survival rate, root mass, and ginsenoside profile. While morphological variations were not quantitatively evaluated, it is generally observed in field practice that wild WSG tends to exhibit more irregular and asymmetrical root structures, denser epidermal layers, and slower growth due to prolonged exposure to natural stressors. In contrast, roots cultivated under controlled environments are typically more uniform in shape, smoother in appearance, and larger in biomass. These tendencies were not formally assessed in this study and are mentioned here only as qualitative background.

Regarding cost, no formal economic analysis was performed. However, based on current market data and cultivation practices, field-cultivated ginseng (FCG; 4–6 years) is typically priced at 30,000–70,000 KRW/kg, while wild-simulated ginseng aged seven years or more is valued at 200,000–250,000 KRW/kg. The post-treated WSG presented in this study is expected to fall between these two categories in terms of cost, requiring more resources than FCG but significantly less than wild cultivation. At the same time, it retains key pharmacological compounds—such as Re, Compound K, and Rg3—that are usually associated with higher-value ginseng products. To reflect this context, we have added the following statement to the Conclusion section (Page 6, Lines 139-147):

“… The post-treated WSG evaluated in this study occupies a distinct position between conventional field-cultivated and wild-simulated ginseng. It shows improved root growth and preservation of pharmacologically relevant ginsenosides under a shorter cultivation window. While the production cost is expected to be higher than that of FCG, it remains significantly lower than that of long-term WSG cultivation. This balance of cost and efficacy may offer a practical and scalable alternative for premium ginseng markets.

Achieving significant growth within five months, high survival rates, and desirable pharmacological properties suggests potential economic benefits. This method could improve the commercial value of previously unmarketable WSG while ensuring consistent quality standards.”

---

## [Editor Report · Decision Letter 1]

Post-harvest Treatment of Wild-simulated Ginseng under Climate-smart Environmental Conditions

PONE-D-25-14014R1

Dear Dr. Kim,

We’re pleased to inform you that your manuscript has been judged scientifically suitable for publication and will be formally accepted for publication once it meets all outstanding technical requirements.

Kind regards,

Chun-Hua Wang

Academic Editor

PLOS ONE

---

## [Editor Report · Acceptance letter]

PONE-D-25-14014R1

PLOS ONE

Dear Dr. Kim,

I'm pleased to inform you that your manuscript has been deemed suitable for publication in PLOS ONE. Congratulations! Your manuscript is now being handed over to our production team.

Kind regards,

on behalf of

Dr. Chun-Hua Wang

Academic Editor

PLOS ONE